# Photodynamic Therapy-Induced Anti-Tumor Immunity: Influence Factors and Synergistic Enhancement Strategies

**DOI:** 10.3390/pharmaceutics15112617

**Published:** 2023-11-11

**Authors:** Wenxin Chou, Tianzhen Sun, Nian Peng, Zixuan Wang, Defu Chen, Haixia Qiu, Hongyou Zhao

**Affiliations:** 1School of Medical Technology, Beijing Institute of Technology, Beijing 100081, China; 3120221957@bit.edu.cn (W.C.); 3120211988@bit.edu.cn (T.S.); pengnian@bit.edu.cn (N.P.); defu@bit.edu.cn (D.C.); 2Department of Laser Medicine, the First Medical Center, PLA General Hospital, Beijing 100853, China; 18352086689@163.com

**Keywords:** photodynamic therapy, innate immunity, specific immunity, influence factors, combination therapy

## Abstract

Photodynamic therapy (PDT) is an approved therapeutic procedure that exerts cytotoxic activity towards tumor cells by activating photosensitizers (PSs) with light exposure to produce reactive oxygen species (ROS). Compared to traditional treatment strategies such as surgery, chemotherapy, and radiation therapy, PDT not only kills the primary tumors, but also effectively suppresses metastatic tumors by activating the immune response. However, the anti-tumor immune effects induced by PDT are influenced by several factors, including the localization of PSs in cells, PSs concentration, fluence rate of light, oxygen concentration, and the integrity of immune function. In this review, we systematically summarize the influence factors of anti-tumor immune effects mediated by PDT. Furthermore, an update on the combination of PDT and other immunotherapy strategies are provided. Finally, the future directions and challenges of anti-tumor immunity induced by PDT are discussed.

## 1. Introduction

Photodynamic therapy (PDT) utilizes an administered photosensitizer (PS) activated by light to achieve localized cytotoxicity for treatment of various indications (Figure 1). At present, PDT has been proven to be an effective strategy in various cancer treatment, including cervical cancer, skin cancer, nasopharyngeal carcinoma, etc. [1,2]. There are three mechanisms of PDT-induced tumor destruction. Firstly, ROS generated by PDT directly kill the primary tumor cells. Secondly, ROS disrupt the tumor vascular system by inducing the release of vasoconstrictors and the form of blood clots. Finally, the immune system is activated by PDT, which can not only eliminate primary tumors, but also effectively destroy metastatic lesions through the activation of T cells [3]. However, the PDT-mediated immune effects are influenced by several factors, such as the localization of PSs in cells, PS concentration, fluence rate of light, oxygen concentration, and the integrity of immune function. Therefore, comprehensive consideration of these factors to achieve the optimal outcome of PDT is important for the physician to make a treatment protocol. In addition, the combination of PDT with immune checkpoints, DC vaccines, and chemotherapy have been reported to effectively improve the immune effects of PDT [4,5]. Herein, we first analyze the mechanism and influence factors of anti-tumor immunity mediated by PDT. Then, the recent advances on combination of PDT with other immunotherapies are also summarized. Finally, the future direction and challenges of anti-tumor immunity induced by PDT are discussed.

## 2. Immunological Effects of PDT

Many studies have demonstrated that innate and specific immunity can be activated by PDT. Tumor cell necrosis induced by PDT is frequently accompanied by an acute inflammatory response and infiltration of inflammatory cytokines, which can trigger innate immunity [6]. Meanwhile, immunogenic cell death (ICD) induced by PDT leads to the release of damage-associated molecular patterns (DAMPs) and the activation of specific immune response [7].

### 2.1. Activation of Innate Immunity

The mechanism of PDT-mediated innate immunity has been widely studied, which mainly include three parts [8,9,10]. Firstly, nuclear factor κB (NF-κB) and activator protein 1 (AP-1) in the tumor cells can be activated after PDT and contribute to the activation of acute inflammatory response [11,12]. Secondly, the activation of inflammatory signaling pathway results in the secretion of inflammatory cytokines and chemokines, such as interleukin (IL), tumor necrosis factor (TNF), and interferon (IFN) [13]. Interleukin-1β (IL-1β) plays an important role in neutrophils infiltration of tumors. The increased activation of neutrophils is an essential symbol for innate immune response induced by PDT [14]. Kousis et al. reported that the activation and proliferation of CD8^+^ T cells is influenced by exhaust of neutrophils [15]. It is widely known that CD8^+^ T cells can preferentially attack tumor cells by recognizing antigens. Thirdly, the activation of the complement system is conducive to clearance of tumor cells after PDT. Stott et al. have demonstrated that C3, C5, and C9 have significantly increased in Lewis lung carcinoma (LLC) cells after Photofrin-PDT [16]. Cecic et al. also discovered that the cure rate of LLC tumors after PDT can be influenced by complement antagonists of C3aR or C5aR [17]. These studies suggested that the complement system is a pivotal part of PDT-mediated anti-tumor immunity.

### 2.2. Activation of Specific Immunity

PDT stimulates specific immune effects by inducing the immunogenic death (ICD) of tumor cells. ICD is a form of cell death to activate immune system, which provokes specific immune response by eliciting danger signals released from dying tumor cells. The occurrence of ICD is one precondition of anti-tumor immunity. Therefore, the development of ICD inducers has become a research hotspot. PSs are considered an efficient ICD inducer under irradiation, especially Hypericin (Hyp) (Figure 1, Compound **1**), which can target endoplasmic reticulum (ER) [18] and trigger ER stress by producing ROS, resulting in the release of a variety of damage-associated molecular patterns (DAMPs) and the activation of specific immune response [19,20,21].

ER stress is an adaptive response of cells to the over-accumulation of damaged proteins. ER receptors are activated when ER is under stress. In order to relieve ER stress and achieve self-help, signaling pathways such as PERK, ATF6 and IRE1 are activated [22]. Among them, PERK is an important regulatory molecule in immunogenicity. It can incite the release of DAMPs such as calreticulin (CRT), high-mobility group protein 1 (HMGB1), and ATP [23]. DAMPs are a kind of dangerous signal released into the extracellular space during tumor cell death [24]. The characteristics of PSs known to trigger ICD and their immune activation effects are summarized in Table 1.

The process of anti-tumor immune activation of PDT is briefly introduced as follows. Firstly, CRT, as an “eat me” signal for APCs, is the most important DAMP that triggers ICD. It can be translocated to the outer surface of the plasma membrane of dying tumor cells after PDT, and then recognized by the low-density lipoprotein receptor-related protein 1 (LRP1) receptor on dendritic cells (DCs). Then, DCs become matured and boost anti-tumor immune response by sparking antigen presentation [25,26,27]. Extracellular ATP, as a “find me” signal, can bind to the purinergic receptor P2Y2 (P2Y2R) and purinergic receptor P2X7 (P2X7R) on DCs. It is worth mentioning that ATP can attract the recruitment of monocytes by recognizing P2Y2 receptors, as well as promote formation of the inflammasome and secretion of inflammatory stimulating factors by combining P2X7 receptors [28]. Additionally, heat shock protein (HSP) and HMGB1 are also important DAMPs which can be triggered by PDT. HSPs can effectively promote DC maturation and stimulate T cells by binding to Toll-like receptor 2 (TLR2) and Toll-like receptor 4 (TLR4) [29,30].

Overall, PDT, as an ICD inducer, generates ROS in tumor cells to stimulate massive exposure of DAMPs, which in turn promotes maturation of DCs and activation of cytotoxic T lymphocytes. Finally, the activated DCs and T cells mediate patient-specific immune effects for the elimination of primary and metastatic lesions (Figure 2). The existing PSs capable of triggering the ICD effects are summarized and listed in the Table 2. The structural formulas of classic PSs are shown in Figure 1.

**Table 1 pharmaceutics-15-02617-t001:** Common DAMPs and their functions.

	DAMP	PRR Receptor	Function	References
1	CRT	LRP1 (CD91)	As a pro-phagocytic signal and promoting antigen presentation	[25,26]
2	ATP	P2RX7	Activate inflammatory bodies and promote the secretion of inflammatory factors	[28,31]
		P2RY2	Attract recruitment of monocytes	
3	HMGB1	TLR2, TLR4, TLR9	Promote DC maturation (especially its metastasis to lymph nodes) and activate T cells	[28,30]
4	HSP70	TLR2, TLR4	Induce DC expression and maturation, and promote cytokine release, especially IL-12 and TNF-α	[32,33,34]
	HSP90			
5	Annexin A1	FPR1	Help DC move to dying cells	[35]
6	CpG DNA	TLR9	Expression of high levels of MHCII and costimulatory molecules (CD80, CD86) and production of IL-12, interleukin, and other cytokines to promote DC maturation and activation	[36,37]
7	CXCL10	CXCR3	Induction of DC activation and T cell infiltration	[38]
8	ExRNA	TLR3	Release TNF-α, IL-1β, or IL-6 and other inflammatory cytokines	[39,40,41]
9	dsDNA	TLR3, RIG-I	Promote the expression of proinflammatory cytokines type I IFN, etc.	[42]
10	dsRNA	TLR3	Promote the expression of proinflammatory cytokines type I IFN, etc.	[43]
11	Type I IFNs	IFNAR1/IFNAR2	Enhance the function of CTL and NK cells and promote the secretion of CXCL10	[44,45,46]
12	ssRNA	TLR7, TLR8	Promote the release of other DAMPs, release cytokines, and promote DC maturation	[47,48]

CRT: Calreticulin; HMGB1: High-mobility group protein box 1; HSP: Heat shock protein; DC: Dendritic cell; TNF: Tumor necrosis factor; IFN: Interferon; IL: Interleukin; MHC: Major histocompatibility complex; CTL: Cytotoxic T lymphocyte.

**Table 2 pharmaceutics-15-02617-t002:** PS-induced ICD.

Photosensitizer	Cell Line	Cell Death Type	Subcellular Localization	DAMP	Immunological Effects of Tumor Cells In Vitro	Immunological Effects of Tumor Cells In Vivo	Reference
Indocyanine green (Figure 1, Compound **9**)	CT26	Apoptosis	ER	CRT	N/D	Maturation of DCs; CD8^+^T cells ↑; TNF-α, IFN-γ ↑; Tregs cells ↓;	[49]
	B16	N/D				Maturation of DCs (CD11c^+^/CD80^+^/CD86^+^↑); CD4^+^ T cells, CD8^+^ T cells ↑; IL-6 ↑, TNF-α, IFN-γ ↑; Tregs cells ↓;	
TCPP-T^ER^	4T1	N/D	ER	CRT, HMGB1	Maturation of DCs (CD80^+^CD86^+^ ↑); IL-12, TNF-α ↑;	CD8^+^ T cells ↑; IL-12, TNF-α, INF-γ ↑;	[50]
Hypericin (Figure 1, Compound **1**)	T24	Apoptosis	ER	CRT, ATP	Phenotypic maturation of DCs (MHC II, CD80^+^, CD83^+^ and CD86^+^ ↑);	DC phenotype maturation (CD80^+^, CD83^+^, CD86^+^, MHC II ↑); IL-1β ↑, IL-10 ↓;	[18]
5-aminolevulinic acid (Figure 1, Compound **6**)	PECA	Apoptosis	ER	CRT, HSP70, and HMGB1	Phenotypic maturation of DCs (CD80^+^, CD86^+^ and MHC II↑); IFN-γ, IL-12 ↑;	N/D	[51]
Porphyrazines (Pz I and Pz III)	MCA205	Apoptosis	Pz-I: GA and Lys	ATP, HMGB1	Maturation of DCs (CD80^+^, CD86^+^ ↑);	N/D	[52]
		Ferroptosis, Necrosis	Pz-III: ER and Lys				
Verteporfin (Figure 1, Compound **7**)	CT26	Apoptosis, Necrosis	N/D	CRT, HSP70, and HMGB1	Maturation of DCs (CD11c^+^CD40^+^CD86^+^ ↑);	Phenotypic maturation of DCs (CD86^+^↑); CTL ↑; IFN-γ ↑; Tregs ↓;	[53]
TPE-PR-FFKDEL	4T1	N/D	ER	CRT, ATP, HMGB1, and HSP70	N/D	DC phenotype maturation (CD80^+^CD86^+^ ↑); CD8^+^ T cells, NK cells ↑;	[54]
Photosens (Figure 1, Compound **10**)	GL261, MCA205	Apoptosis, Ferroptosis	Lys	CRT, HMGB1, and ATP	Maturation of DCs (CD86^+^ ↑); IL-6 ↑;	N/D	[55]
Photodithazine	GL261, MCA205	Apoptosis	ER and GA	CRT, HMGB1, and ATP	DC phenotype maturation (CD86^+^ ↑); IL-6 ↑;	N/D	[55]
Photofrin (Figure 1, Compound **2**)	C-26	Apoptosis, Necrosis	N/D	HSP	Maturation of DCs (IL-2 ↑);	CD8^+^ T cells, NK cells ↑;	[56]
Cu-TBP nMOF	B16F10	Apoptosis	N/D	CRT	N/D	Maturation of DCs (CD11^+^ ↑); IFN-β, CD4^+^ T cells and CD8^+^ T cells ↑;	[57]
Chlorin e6 (Figure 1, Compound **5**)	4T1	Apoptosis	N/D	CRT	Phenotypic maturation of DCs (CD80^+^, CD86^+^ ↑);	DC phenotype maturation (CD86 ↑); CD8^+^ T cells, CD4^+^ T cells ↑;	[58]
Chlorin e6 (Figure 1, Compound **5**)	B16	Apoptosis	N/D	CRT, HSP90, HMGB1, and ATP	MI macrophage activation (GBP5, iNOS and MHC-II ↑); IFN-β ↑;	N/D	[59]
Indocyanine green (Figure 1, Compound **9**)	MC38	Apoptosis	N/D	CRT, HSP70, and ATP	Maturation of DCs (CD86^+^ ↑); IL-12-p40 ↑;	Significantly inhibited tumor growth;	[60]
	CT26						
Pyrolipid	4T1	Apoptosis, Necrosis	N/D	CRT	N/D	TNF-α, IL-6 and IFN-γ ↑; B cells, CD8^+^ T cells ↑; Significantly inhibited tumor growth;	[61]
Chlorin e6 (Figure 1, Compound **5**)	4T1	Apoptosis	N/D	CRT	DC phenotype maturation (CD80^+^CD86^+^ ↑);	DC phenotype maturation (CD80^+^CD86^+^ ↑); CD8^+^ T cells ↑;	[62]
Chlorin e6 (Figure 1, Compound **5**)	4T1	Apoptosis	N/D	CRT, HMGB1, and ATP	Phenotypic maturation of DCs (MHC II, CD86 ↑);	CD8^+^ T cells, CD4^+^ T cells and NK cells ↑;	[63]
Pyropheophorbide	CT26	Apoptosis, Necrosis	N/D	CRT	N/D	TNF-α, IL-6 and IFN-γ ↑;	[64]
Porphyrin (Figure 1, Compound **4**)	CT26	Apoptosis	N/D	CRT, ATP, and HMGB1	N/D	DC phenotype maturation (CD80^+^CD86^+^ ↑); CD8^+^ T cells ↑;	[65]
Chlorin e6 (Figure 1, Compound **5**)	LLC or A549	Apoptosis	N/D	CRT, HSP 90, and HMGB1	MHC I ↑	Ce6-PDT showed excellent anti-tumor efficacy; MHC I ↑;	[66]
Indocyanine green (Figure 1, Compound **9**)	4T1	N/D	N/D	CRT	N/D	Phenotypic maturation of DCs (CD86, CD80 ↑); IL-10 and IFN-γ ↑; CD8^+^ T cells, CD4^+^ T cells and NK cells ↑; TGF-β ↓;	[67]
Core-shell gold nanocages coated with manganese dioxide (AuNC@MnO_2_)	4T1	Apoptosis	N/D	CRT, ATP, and HMGB1	DC phenotype maturation (CD83, CD86 ↑); IL-12 ↑;	Maturation of DCs (CD86^+^ ↑); NK cells, CD8^+^ T cells and CD4^+^ T cells ↑; Treg cells↓;	[68]
Chlorin e6 (Figure 1, Compound **5**)	4T1	Apoptosis, Necrosis	Cytoplasm	CRT, ATP	Maturation of DCs (CD80^+^CD86^+^ ↑);	Significantly inhibited tumor growth; CD4^+^ T cells, CD8^+^ T cells ↑;	[69]
2-(1-hexyloxyethyl)-2-devinyl pyropheophor-bide-a (HPPH)	B16F10	Apoptosis	Endo/Lys then in ER	CRT	N/D	IL-6, TNF-α ↑; CD8^+^ T cells ↑;	[70]
Zinc-phthalocyanine	MC38	Apoptosis	Mitochondria	CRT	N/D	Significantly inhibited tumor growth	[71]
Zinc-phthalocyanine	TC-1	Pyroptosis	Mitochondria	CRT, HMGB1	N/D	Significantly inhibited the growth of primary tumors and metastatic tumors	[72]
IR780	CT26	N/D	Mitochondria	CRT, ATP, HMGB1, and HSP90	Phenotypic maturation of DCs (CD80^+^CD86^+^ ↑);	CD4^+^T cells and CD8^+^T cells ↑; Significantly inhibited the growth of primary tumors and metastatic tumors	[73]
TPE-DPA-TCyP	4T1	N/D	Mitochondria	CRT, ATP, HMGB1, and HSP70	N/D	DC phenotype maturation (CD80^+^ CD86^+^ ↑); Significantly inhibited the growth of primary tumors and metastatic tumors; CD4^+^ T cells, NK cells ↑;	[74]

N/D: Not detected; ER: Endoplasmic Reticulum; GA: Golgi Apparatus; Lys: Lysosomes; CRT: Calreticulin; HMGB1: High-mobility group protein box 1; HSP: Heat shock protein; DC: Dendritic cell; TNF: Tumor necrosis factor; IFN: Interferon; IL: Interleukin; MHC: Major histocompatibility complex; Tregs: Regulatory T cells; ↑: Increase in proportion; ↓: Reduction in proportion.

## 3. Influence Factors of the Anti-Tumor Immunity Induced by PDT

The three elements of PDT are PSs, light, and oxygen, which determine the efficacy of PDT and the death mode of cancer cells. Hence, the immunity induced by PDT relies on several factors, including the localization and dose of PSs, light fluence, and concentration of oxygen in tumors. Understanding and regulation of these influencing factors are important to improve the anti-tumor immune effects of PDT.

### 3.1. Localization of PSs

The immune effects induced by PDT is highly related to the intracellular localization of PSs. The mitochondria, ER, Golgi apparatus, and lysosomes are the binding sites of PSs in the cells [75]. Different PSs mainly bind to the different organelles. Among these organelles, ER maintains the intracellular calcium homeostasis and protein folding, which plays a central role in immunogenic cell death [3,66]. Many studies have examined the relationship between ER stress and the efficiency of induction of ICD. Garg et al. found that Hyp-PDT could trigger CRT exposure, but not Photofrin-PDT under the same conditions because Hyp is primarily located in the ER, whereas Photofrin (Figure 1, Compound **2**) is dispersed within the cells and binds to the ER, mitochondria, and Golgi apparatus [76]. Brodin [77] and Alzeibak [78]’s studies highlight the role of ER stress response for incited ICD. Therefore, ER targeting of PSs is a decisive factor for the anti-tumor immunity of PDT. Currently, a few PSs can target the ER have been reported, including mTHPC (Foscan) (Figure 1, Compound **3**) [79], Benzoporphyrin derivative (BPD) [80], and Hyp [81]. However, there is still a lack of ER-targeted PSs in the clinic. Turubanova et al. developed a new PS porphyrins III (Pz III) [52], which targets to ER and causes ER stress and DAMP release under 630 nm laser irradiation (Figure 3A). Particularly, there was a significant increase in HMGB1 and ATP release (Figure 3B,C). In addition, the activation and maturation of BMDCs were induced by MCA205 dying cells treated with Pz III-PDT (Figure 3D). Similarly, in C57BL/6J mice model, the inhibitory effects on tumor growth of the Pz III-PDT group were more pronounced than PBS-injected group (Figure 3E). These results indicated that tumor dying cells after Pz III-PDT can be immunogenic, which would contribute to the stimulation of an adaptive immune response.

Li et al. designed a new ER-targeted PS, TPE-PR-FFKDEL, which has been proven to be effective in inducing ICD [54]. The exposure of CRT and the release of HSP70, HMGB1, and ATP were detected in 4T1 cells after TPE-PR-FFKDEL-PDT. This marked preponderance of released DAMPs can be attributed to the ability of TPE-PR-FFKDEL targeting ER. Furthermore, TPE-PR-FFKDEL-PDT observably led to higher numbers of CD8^+^ T cells and NK cells in the spleen of mice, indicating the effective provoking of specific and innate immunity. All these results indicate that designing ER-targeted PSs are an effective strategy to induce ICD and adaptive anti-tumor immune response.

In addition, transporting PSs with nanoparticles (NPs) to the subcellular organelles is another strategy for improving immunity of PDT. Zhang et al. obtained Ce6-IMDQ NPs via self-assembly, which combined chlorin e6 (Ce6) (Figure 1, Compound **5**) with TLR7 agonists (IMDQ) (Figure 4A) [58]. Upon 660 nm laser irradiation, Ce6-IMDQ NPs could induce tumor cell death and antigen release, which effectively upregulated CRT exposure and induced DC maturation (Figure 4D). Flow cytometric analyses confirmed that the Ce6-IMDQ NPs group induced higher level of CD8^+^ T cells and CD4^+^ T cells in the spleen and the distant tumors. In particular, the numbers of CD8^+^ T cells increased from 7.85% to 14.14% (Figure 4E). Therefore, the stronger anti-tumor effects of PDT were elicited by Ce6-IMDQ NPs through promoting the infiltration of cytotoxic T lymphocytes. Similar results were also obtained in primary and distant animal tumor models. Tumor growth was more effectively inhibited in the Ce6-IMDQ NPs treatment group compared to the other treatment groups (Figure 4B,C).

Deng et al. [50] developed new NPs, Ds-sP/TCPP-T^ER^, which consist of reduction-sensitive Ds-sP NPs (PEG-s-s-1,2distearoyl-sn-glycero-3-phosphoethanolamine-N-[amino(polyethylene glycol)-2000] NPs) and an efficient ER-targeting PS TCPP-T^ER^ (Figure 5A). Ds-sP/TCPP-T^ER^ NPs could selectively accumulate in the ER. Upon 670 nm laser irradiation, ICD was induced by the expression of ecto-CRT and HMGB1 (Figure 5B,C). The immunogenic effects of Ds-sP/TCPP-T^ER^-PDT were further demonstrated in vivo (Figure 5D–F). Therefore, PSs modified by NPs with ER-targeting function can effectively boost immune effects induced by PDT and show extraordinary performance in eliminating primary and metastatic tumors.

In addition to the ER, other organelles targeted by PSs can also induce the release of DAMPs under laser irradiation. For instance, mitochondria are the primary site for aerobic respiration and participate in energy supply, signal transduction, and apoptosis [82]. Wang et al. have demonstrated that 5-aminolevulinic acid (5-ALA) (Figure 1, Compound **6**), as a mitochondrial-targeted PS, was able to substantially increase the expression of CRT, HSP70, and HMGB1, followed by maturation of DCs [51]. Another important organelle is the lysosome which, as the main recycling organelle of the cell, is associated with digestion and autophagy [83]. When cells are damaged, the lysosomal membrane will rupture to trigger cell death by initiating the release of tissue proteases and ROS [84]. Therefore, lysosome-mediated cell death in various forms has received wide attention in recent years. Turubanova et al. [55] found that a PS-targeting lysosome named Photosens (Figure 1, Compound **10**) can induce CRT exposure and HMGB1 and ATP release after 635 nm laser irradiation. MCA205 tumor growth was significantly inhibited when the mice were injected with the dead MCA205 cells treated by Photosens-PDT. These results indicated that the specific immune response was successfully triggered. In a word, mitochondria play a more important role in cell death, especially in apoptosis induced by PDT than lysosomes because pro-apoptotic cytochrome c is located in the mitochondria. However, lysosomes have a greater contribution in the immune activation of PDT.

### 3.2. Dose of PSs

In addition to subcellular localization, the dose of PS also influences PDT-mediated antitumor immunity. For example, PDT-treated of SCC cells with 1 µM OR141 can evoke more release of HSP90 and HMGB1 compared to 10 µM OR141, which indicates that a stronger ICD is induced by low dose of OR141 [85]. It was further confirmed by in vivo experiments that the growth of SCC tumors is more efficiently inhibited after treatment with 4 mg/kg OR141 in comparison with 40 mg/kg. In addition, Morais et al. showed that the AlPc (Figure 1, Compound **8**) can cause the release of HMGB1 at concentrations of 4.3 nM, 7.8 nM, and 12.2 nM under the same dose of laser irradiation, but the release of ATP only happened at a concentration of 12.2 nM [86]. It is hypothesized that this is due to a non-linear correlation between PSs dose and ICD effects. Therefore, the dose of PSs is an important parameter for PDT-induced ICD effects. The dose of PSs should be selected to provide maximum activation of the immune effects while minimizing damage to normal cells.

### 3.3. Light Fluence Rate

The laser energy used in PDT is another factor that may affect the anti-tumor immune response. The excitation of PSs is influenced by the energy density and power density of the laser. These two parameters are capable of influencing PDT-induced inflammatory response and tumor suppression by altering the oxygen concentration in tumor tissues. Henderson et al. compared the inflammatory response induced by different energy density (48 J/cm^2^, 128 J/cm^2^) and power density (14 mW/cm^2^, 112 mW/cm^2^). They revealed that low power density and low energy density could trigger the stronger inflammatory response [87]. However, this PDT regimen had poor local tumor suppression. In contrast, the strongest anti-tumor effects as well as the smallest inflammatory response was obtained by high energy density and low power density. In a subsequent study, it was found that high energy density (128 J/cm^2^) and high power density (224 mW/cm^2^) displayed the worst tumor suppression effects [88], because the oxygen in the tumor tissues were seriously depleted by the higher light energy. The sharp decrease in oxygen concentration within tumor tissues may lead to remarkable reduction in ROS production, which ultimately severely limited the therapeutic efficacy of PDT. To solve this problem, a two-step combination therapy was designed, which combined low energy density (48 J/cm^2^) and low power density (14 mW/cm^2^) with high energy density (132 J/cm^2^) and low power density (14 mW/cm^2^) [89]. This combination therapy has significant advantages in enhancing anti-tumor immunity, especially in promoting the recruitment of CD8^+^ T cells and inhibiting the growth of murine colon and mammary tumors. Therefore, the treatment protocol formulated with optimal light parameters not only significantly inhibits the growth of primary tumors, but also effectively activates anti-tumor immunity. 

### 3.4. Oxygen Content

Recently, many studies have revealed the association between specific immunity of PDT and oxygen content in tumors [2]. The tumors will develop a hypoxic microenvironment due to the continuous consumption of oxygen during PDT, which can cause the immune suppression of tumors. Therefore, alleviating the hypoxic state of tumors is one of the effective strategies to improve the anti-tumor efficacy of PDT.

Strategy one: constructing a hypoxia-reverse nanosystem to alleviate tumor hypoxia during PDT. As an example, Li et al. developed novel NPs with double ER targeting function [49]. The NPs consisting of indocyanine green (ICG) (Figure 1, Compound **9**) conjugated-hollow gold nanospheres (named: fAL-ICG-HAuNS) and oxygen-delivering hemoglobin (named: FAL-Hb lipo) are endowed with the ability to target ER after modification with pardaxin (FAL) peptides (Figure 6A). Under hypoxic conditions, the ICG-HAuNS plus Hb-lipo group resulted in approximately 46% cell death compared to only 20% in free ICG, HAuNS, and ICG-HAuNS groups (Figure 6B). Correspondingly, in mice with B16 tumors, fAL-ICG-HAuNS plus FAL-Hb lipo treatment dramatically reduced the proliferation rate of tumors and effectively extended the life of mice. It can be speculated that the ICG-HAuNS plus FAL-Hb lipo treatment can induce 33.1% DC maturation (MHC I^+^/MHC II^+^) in the tumors and activate specific effector cells (CD8^+^ T cells ^high^) (Figure 6C). Therefore, this double ER-targeting strategy was demonstrated to induce anti-tumor immunity and eliminate primary tumors.

Liang et al. [68] designed a PDT oxygen enhancement generator, which is a core gold nanocage with a shell of manganese dioxide (named: AuNC@MnO_2_, AM) (Figure 7A). Under laser stimulation, a high level DAMP release and DC maturation (CD83 ^high^, CD86 ^high^) can be triggered by AM-PDT. The immunogenic effects of AuNC@MnO_2_-PDT were further demonstrated in vivo to effectively inhibit the growth of primary tumors and lung metastases in 4T1-bearing mice (Figure 7B–D). Therefore, the results indicated that oxygen-enhanced PDT serves as a potent pathway for triggering ICD, which dramatically enhances PDT-mediated immune effects by promoting the recruitment of effector T cells.

In addition to improving oxygen delivery efficacy, developing PSs with type I mechanisms to decrease oxygen consumption of PDT is another strategy to improve immunological effect. Compared to the type II mechanism, the type I mechanism of PDT has less oxygen consumption, thereby alleviating the anoxic microenvironment of tumors during PDT [90]. Huang et al. developed a self-degrading conjugated polyelectrolyte (named: CP^+^) consisting of aggregation-induced emission (AIE) and imidazole units, which can effectively generate O_2_^•−^ by a type I mechanism [91]. At the same time, CP^+^ was loaded with the immunoadjuvant cytosine-phosphate-guanine (CpG). Compared with other groups, the CP^+^-CpG NPs group resulted in a significant increase in the proportion of CD4^+^ T cells and CD8^+^ T cells, which indicated successful activation of the immune effects and inhibition of primary and metastatic tumors. Chen et al. designed a set of acridinium derivatives with extended D-π-A systems (named: IMA, QMA, MAMA, BMA, and BAMA) [92]. BAMA-PDT treatment showed significant anti-tumor efficacy in 4T1 tumor models. These results showed that the infiltration of CD8 ^+^ T cells increased 4.8 fold and the proportion of Treg cells decreased in the BAMA-PDT group compared with the PBS group in 4T1 tumors. Therefore, the type I mechanism of PDT can mitigate the decreased anti-tumor immune effects caused by hypoxia.

### 3.5. Immune System

In addition to the above effectors, the integrity of the patient’s immune system function is crucial for long-term inhibition of tumor growth. Korbelik et al. found that the majority of tumors with neutrophil-depleting treatment by Photofrin (Figure 1, Compound **2**)-PDT recurred at 2–3 weeks, although the immediate therapeutic effect was unaffected significantly [93]. This may be due to that immune deficiency commonly accompanies abnormal function of immune cells, such as T lymphocytes and B lymphocytes. Cytotoxic T lymphocytes are integral parts of the adaptive immune system which play an important role in the direct killing of tumor cells [94]. It was confirmed by further experiments that after depletion of CD8^+^ T cells, PDT treatment reduced the tumor cure rate from 100% to 50%, with a more significantly impact than CD4^+^ T cells depletion. In addition to specific immune cells, the activation of innate cells are also essential for immune effects [95]. Korbelik et al. detected that the growth of tumors in mice with severe combined immune deficiencycould not be effectively inhibited by PDT [96]. And, with the depletion of natural killer cells (NK cells), the anti-tumor effect of PDT in mice of immune deficiency was noticeably diminished. Therefore, the integrity of immune system plays a critical role in PDT-mediated immune effects.

## 4. PDT Combined with Other Therapies

Although it has been demonstrated that PDT can trigger immune effects in vivo, the efficacy is often limited by negative factors such as the tumor hypoxic microenvironment and tumor immune escape [97]. At present, PDT in combination with DC vaccines, immune checkpoint inhibitors, chemotherapy, and radiotherapy has been proposed to improve the anti-tumor effect [61,98,99].

### 4.1. PDT Combines with DC Vaccines

Dendritic cells are powerful antigen-presenting cells that can efficiently present antigens to T and B lymphocytes [100]. Many studies have suggested that injecting DC vaccines into patients can induce T lymphocyte activation and IL-12 expression [101,102]. The ICD induced by PDT can achieve the establishment of DC vaccines [58]. Garg et al. produced a DC vaccine based on Hyp-PDT, which can promote the infiltration of T lymphocytes (CD8^+^ T cells, CD4^+^ T cells) and reduce the number of Treg cells in GL261 tumors [103]. The results indicate that Hyp-PDT-based DC vaccines can lead to the alleviation of the immunosuppressive microenvironment. Zhang et al. found that DC vaccines made from ALA-PDT can enhance the activity of effector T cells (CD8^+^ T cells^high^, CD4^+^ T cells^high^) and promote the release of cytokines (IL-12^high^, IFN-γ^high^) in mice with PECA tumors [104]. The same results were also found in another study with the H22 vaccine prepared by hematoporphyrin monomethyl ether (HMME)-PDT (Figure 1, Compound **11**). Compared with the control group, the H22 vaccine group induced an increased number of CD4^+^CD8^+^CD19^+^ T cells and significantly inhibited tumor growth [105]. Therefore, the DC vaccine produced by PDT can effectively ameliorate the immune suppression microenvironment, thereby inhibiting tumor growth and enhancing survival rate.

### 4.2. PDT Combines with Immune Checkpoint Inhibitors

The inhibitory effects of the immune system on tumors are affected by receptors and ligands on tumors and immune cells. These receptors and their ligands are called immune checkpoints, which will be upregulated in tumor microenvironment to resist anti-tumor immunity [106]. Therefore, the blockade of immune checkpoint can improve the recognition of the immune system to tumor cells. In order to enhance the therapeutic efficacy of immune checkpoint blockade (ICB), the combination of immune checkpoint inhibitors-PDT has been extensively studied [107]. The details of the PSs and the types of immune checkpoint inhibitors are summarized in Table 3. As an example, Duan et al. loaded PS pyrolipid in the Zn-pyrophosphate (ZnP) NPs (Figure 8A), ZnP@pyro-PDT treatment sensitizes tumors to immunotherapy mediated by PD-L1 antibodies [61]. The combination of ZnP@pyro-PDT with PD-L1 checkpoint blockade therapy can increase the infiltration of T cells (CD8^+^ T cells^high^, CD4^+^ T cells^high^) (Figure 8D) and significantly eradicate the growth of primary tumors and prevent lung metastases in 4T1 mice models (Figure 8B,C).

Choi et al. developed the NPs LT-NPs, which is self-assembled from the PS verteporfin (VPF) (Figure 1, Compound **7**), cathepsin B-specific cleavable peptide (FRRG) and doxorubicin (DOX) conjugates (Figure 9A) [53]. As a visible-light-triggered prodrug, the NPs (LT-NPs) can transform the tumor immune suppressive microenvironment into one with high immunogenicity, thereby enhancing checkpoint blocking immunotherapy. In the model of bilateral CT26 tumor-bearing mice, the combination of LT-NPs and PD-L1 checkpoint blockade therapy had shown significant advantages in inhibiting tumor growth. The combined therapy completely regressed the tumors within 100 days, and effectively alleviated the lung metastasis of tumors (Figure 9B,C). It is worth noting that ICB combined with PDT treatment is usually accompanied by multiple injections of immune checkpoint inhibitors and even multiple laser irradiation [117].

### 4.3. PDT Combines with Chemotherapy

Chemotherapy (chem), surgery, and radiotherapy are known as the three major treatment methods for tumors. Some chemotherapy drugs, such as doxorubicin [118], idarubicin [119], and oxaliplatin [120], have been shown to cause immunogenic cell death. Therefore, chem/PDT-induced ICD can trigger a systemic anti-tumor immune affect and reverse the immunosuppressive microenvironment of the tumors. The relevant studies on chem/PDT are summarized in Table 4. Yao et al. synthesized a self-cascading unimolecular prodrug (named AIE-pep-DOX), which is formed by coupling doxorubicin and the aggregation-induced emission PS to a caspase-3 response peptide [121]. Compared to the control group, in 4T1 murine models, the AIE-pep-DOX group can alleviate the tumor suppressor microenvironment by increasing the percentage of cytotoxic T cells (CD8^+^ T cells^high^, Tregs^low^), thereby inhibiting tumor growth. Hypoxia and high GSH are typical features of the tumor-suppressing microenvironment, which seriously affects the therapeutic efficacy of PDT by reducing ROS production. Wang et al. constructed a multicomponent supramolecular nanomedicine (named NP_Ce6/Pt_) which combined β-cd modified chlorine e6, cisplatin and hydrophilic poly (oligoethylene glycol) methacrylate [122]. NP_Ce6/Pt_ can be used for GSH depletion, that is, triggering azo cleavage under hypoxic conditions. Therefore, NP_Ce6/Pt_ induced more DAMP release, further enhanced immunogenicity, and improved the DC maturation rate from 5.2% to 28.8% in 4T1 tumor. The percentage of CD8^+^ T cells and CD4^+^ T cells increased 1.35 fold and 1.26 fold, respectively, in the NP_Ce6/Pt_ group. In conclusion, the chem–PDT combination strategy will shed light on enhancing the immunogenicity of the tumors and improving the efficacy of PDT.

## 5. Conclusions and Perspectives

PDT not only kills primary tumors but also inhibits metastatic tumors by mediating both innate and specific immunity of the body. PDT-induced immune response is influenced by the localization and dosage of PSs in cells, parameters of light, concentration of oxygen in tumors, and integrity of immune function. To figure out the relationship between these influencing factors and the immune effect of PDT is very important, which assists physicians to make an optimum treatment plan to patients. Therefore, we summarized and analyzed the influencing factors of PDT-induced anti-tumor immunity in this review. Furthermore, the future direction and challenges of anti-tumor immunity induced by PDT are discussed.

Firstly, ER plays a very important role in inducing immunogenic death of tumor cells. Therefore, the development of ER-targeting PSs is an important option to enhance the anti-tumor immunity of PDT. The biosafety of new PS- or NP-packaged PSs should be further evaluated by sufficient pharmacokinetic research. Secondly, the hypoxia of tumors severely inhibits the PDT-mediated ICD effect. Therefore, the development of type I PSs with oxygen non-dependence is another strategy to improve the anti-tumor immunity of PDT. Improvement of the synthetic efficiency and quantum yield of type I PSs is an urgent problem to be solved. Finally, the combination of PDT and other immunotherapy strategies is a promising direction of improvement of outcome of PDT. Although the synergistic immune effect of combination therapies has been demonstrated by many preclinical studies, only a few combined strategies have progressed to the clinical stage. Meanwhile, the safety of the combination therapy also needs to be evaluated. Anti-tumor immunity is the most accurate and effective tumor-killing strategy. We believe that anti-tumor immunity induced by PDT will benefit more patients after comprehensive consideration of the influencing factors, such as improving the anti-tumor immunological effect through combination of PDT with PD1/PDL1, balancing the elimination efficacy to local tumors and the killing efficacy to metastatic tumors of PDT through regulating the dosage of irradiation and PSs.

## Data Availability

Not applicable.

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
