# Peer review of "Photodynamic Therapy-Induced Anti-Tumor Immunity: Influence Factors and Synergistic Enhancement Strategies"

_pharmaceutics, 2023, doi:10.3390/pharmaceutics15112617_

Round 1
Reviewer 1 Report
Comments and Suggestions for Authors
The paper of Chou et al is a complete review related to PDT-mediated antitumor immunity. The main aspects and parameters influencing photoinduced immunity are addressed in this paper. The reference list is complete (120 papers) ranging from 1995 to 2023.
There are some minor concerns.
1. It would be very helpful to provide the Table of content at the beginning of manuscript.
2. § 3.1. Localization. The authors note that there is a lack of ER-localized PSs in clinic. Please mention mTHPC (Foscan), which demonstrated a clear ER localization (par ex Kessel D. 2021; Teiten et al. 2003). Further, please, mention that mitochondria is extremely important in photoinduced cell death, mostly by apoptotic pathway.
As for lysosomes, while they can trigger specific immune response, their contribution to PDT-induced cell death is not so much important. Please note it.
3. A very good passage about PD-L1 checkpoint inhibition supplied with recent references in the Table 1. The references are correct but it would be helpful to mention in the text that anti-PD-L1 therapy implies multiply injections and in many cases multiply irradiations.
Comments on the Quality of English LanguageNo problems with english language
Author Response
Point-by-Point Response to Reviewer’s Comments
NOTE: The texts in italics are the reviewer’s Specific Comments. Our responses are marked in blue. The revised sentences are marked in red in the revised manuscript.
Reviewer 1
General Appreciation:
- It would be very helpful to provide the Table of content at the beginning of manuscript.
Response: We appreciate this suggestion from the reviewer. We have added the table of content at the beginning of manuscript, which is marked in red in the manuscript (Page 1- Page 2)
- 3.1. Localization. The authors note that there is a lack of ER-localized PSs in clinic. Please mention mTHPC (Foscan), which demonstrated a clear ER localization (par ex Kessel D. 2021; Teiten et al. 2003). Further, please, mention that mitochondria is extremely important in photoinduced cell death, mostly by apoptotic pathway.
As for lysosomes, while they can trigger specific immune response, their contribution to PDT-induced cell death is not so much important. Please note it.
Response: We thank the reviewer for these suggestions. We carefully read the recommended references and add them in this manuscript (Reference [36] and [37]). The ER targeted PS, mTHPC (Foscan), has been described in the revised manuscript (Page 6, line 109-110).
“Currently, a few PSs can target the ER have been reported, including mTHPC (Foscan) (Scheme 1, Compound 3) [36], Benzoporphyrin derivative (BPD) [37], and Hyp [38].”
The description about mitochondria and lysosomes has also added. The updated sentences are marked in red in the revised manuscript (Page 10, line 165-167).
“In a word, mitochondria play a more important role in cell death, especially in apoptosis induced by PDT than lysosomes because pro-apoptotic cytochrome c locates in mitochondria. However, lysosomes have more contribution in immune activation of PDT.”
- A very good passage about PD-L1 checkpoint inhibition supplied with recent references in the Table 1. The references are correct but it would be helpful to mention in the text that anti-PD-L1 therapy implies multiply injections and in many cases multiply irradiations.
Response: We appreciate this suggestions from the reviewer. We have revised the manuscript according to the suggestions. The updated sentences are marked in red in the manuscript (Page 16, line 292-294).
“It is worth noting that ICB combined with PDT treatment is usually accompanied by multiple injections of immune checkpoint inhibitors and even multiple laser irradiation [75].”

Reviewer 2 Report
Comments and Suggestions for Authors
The manuscript “Photodynamic Therapy induced antitumor immunity: Influence Factors and synergistic enhancement strategies” by Chouet al. is an interesting work, describing factors that influence the photodynamic therapy such as PS concentration, fluence rate of light, oxygen concentration, and the integrity of immune function. The manuscript is well organized, images are appealing and data are clearly presented. Only few corrections in style and contents need to be done to improve the quality of presentation. Hence, it is suitable for publication after the following minor revisions:
- The format for figure 2 and 3 should be checked since they seem overlapped at the end of page 5. Furthermore, images are well designed but have low quality, making data analysis difficult.
- Place all the captions in the same position: below or above the corresponding figure. In the current state, they are casually placed.
- Authors should focus more on the different PSs and their molecular structures. None of them is shown. Add a figure or include them in a table.
- Paragraph 3.4: present the two strategies as two different subparagraphs, otherwise rephrase and try to better argue the introduction to each strategy.
- Future perspectives are too generic and should be improved, giving examples of upcoming strategies in clinical phase
- At the end of line 27, references are missing. Please add recent and relevant examples of studies in the field of PDT such as: Photodyn Ther. 2023 Sep;43:103644. doi: 10.1016/j.pdpdt.2023.103644; Arch Pharm Res 45(11):1-16 DOI: 10.1007/s12272-022-01414-1; J Eur Acad Dermatol Venereol. 2022 Nov;36(11):e946-e948. doi: 10.1111/jdv.18374.
- English should be revised in many points. For example: in line 83, change “is important” with “are important”; rephrase lines 178-180, 247, 309-310; line 503, check “hypoxia is one of THE reasons”
Comments on the Quality of English LanguageMinor editing of English language required
Author Response
Point-by-Point Response to Reviewer’s Comments
NOTE: The texts in italics are the reviewer’s Specific Comments. Our responses are marked in blue. The revised sentences are marked in red in the revised manuscript.
Reviewer 2
General Appreciation:
- The format for figure 2 and 3 should be checked since they seem overlapped at the end of page 5. Furthermore, images are well designed but have low quality, making data analysis difficult.
Response: We appreciate the suggestions from the reviewer. We have modified the formatting of Figures 2 and 3, and the separation rate and size of all figures have been improved.
- Place all the captions in the same position: below or above the corresponding figure. In the current state, they are casually placed.
Response: We thank the reviewer for helping us point it out. All the captions have been placed below the corresponding figures.
- Authors should focus more on the different PSs and their molecular structures. None of them is shown. Add a figure or include them in a table.
Response: We thank the reviewer for the suggestion. Three schemes have been added to show the chemical structural formula of PSs.
- Paragraph 3.4: present the two strategies as two different subparagraphs, otherwise rephrase and try to better argue the introduction to each strategy.
Response: We really appreciate the suggestion from the reviewer. We've added a description of two strategies, which has been marked in red in the revised manuscript (Page 13, line 230-231).
“Besides of improving oxygen delivery efficacy, developing PSs with type I mechanism to decrease oxygen consumption of PDT is another strategy to improve immunological effect.”
- Future perspectives are too generic and should be improved, giving examples of upcoming strategies in clinical phase.
Response: We thank reviewer for the suggestion. Future perspectives have been revised, which has been marked in red in the revised manuscript. (Page 17, line 334-346).
- At the end of line 27, references are missing. Please add recent and relevant examples of studies in the field of PDT such as: Photodyn Ther. 2023 Sep; 43:103644. doi: 10.1016/j.pdpdt.2023.103644; Arch Pharm Res 45(11):1-16 DOI: 10.1007/s12272-022-01414-1; J Eur Acad Dermatol Venereol. 2022 Nov;36(11): e946-e948. doi: 10.1111/jdv.18374.
Response: We are very grateful to the reviewer for recommending the good references. These papers have been cited in the manuscript (Reference [2], [4], and [5]).
- English should be revised in many points. For example: in line 83, change “is important” with “are important”; rephrase lines 178-180, 247, 309-310; line 503, check “hypoxia is one of THE reasons.”
Response: We thank the reviewer for helping us point them out. We have carefully checked and revised the grammar of the manuscript.
- “is important”has been replaced by “are important”.
- “Therefore, developing an optimal PDT treatment regimen by utilizing different light parameters to not only significantly suppress primary tumor growth, but also efficiently active anti-tumor immunity, which is meaningful for patients.”has been replaced by “Therefore, the treatment protocol formulated with optimal light parameters not only significantly inhibits the growth of primary tumors, but also effectively activates anti-tumor immunity”
- “Garg et al.produced a DC vaccine-based on Hyp-PDT can promote the infiltration of specific effector cells (CD8+ T cells high、CD4+ T cells high) in mice with GL261 tumors and reduce Treg cells” has been replaced by “Garg et al. produced a DC vaccine based on Hyp-PDT, which can promote the infiltration of T lymphocytes (CD8+ T cells, CD4+ T cells) and reduce the number of Treg cells in GL261 tumors.”
- “Despite the large number of preclinical studies have demonstrated the immune-modulating properties of combination therapy”has been replaced by “Although the synergistic immune effect of combination therapies has been demonstrated by many preclinical studies”
- “Second, tumor hypoxia is one of reasons which alleviates the ICD efficiency of PDT.”has been replaced by “Secondly, the hypoxia of tumor severely inhibits the PDT-mediated ICD effect.”

Reviewer 3 Report
Comments and Suggestions for Authors
This review delves into the immunogenic impact of PDT, highlighting its potential to eliminate primary tumors and inhibit metastatic growth. The authors emphasize the role of factors such as photosensitizer location, light settings, tumor oxygen levels, and immune function in shaping PDT-induced responses. They underscore the importance of targeting the ER for immunogenic cell death and address the challenges of managing tumor hypoxia with oxygen-independent photosensitizers.
The review explores the combination of PDT with immunotherapies to improve treatment outcomes, bridging preclinical findings with clinical applications. It serves as a valuable resource for researchers in the field, offering insights into research status, challenges, and potential solutions. While the content is informative and aligns with potential publication standards, a few minor comments have been noted:
1. The inclusion of a Jablonski diagram depicting the PDT action mechanism could enhance comprehension (PMID: 18037279).
2. Further discussions on the combination of PDT with radiotherapy could be beneficial, with consideration of citations from articles like PMID: 24480782, PMID: 28094262, PMID: 34094142, and PMID: 33643804.
3. Typographical errors, such as 'et al.' in line 20, could be replaced with 'etc.'
4. Additionally, typographical errors are observed in Line 119, Line 159, etc.
Author Response
Point-by-Point Response to Reviewer’s Comments
NOTE: The texts in italics are the reviewer’s Specific Comments. Our responses are marked in blue. The revised sentences are marked in red in the revised manuscript.
Reviewer 3
General Appreciation:
- The inclusion of a Jablonski diagram depicting the PDT action mechanism could enhance comprehension (PMID: 18037279).
Response: We appreciate this suggestion from the reviewer. We have added a Jablonski diagram of PDT action mechanism in revised manuscript (Figure 1).
- Further discussions on the combination of PDT with radiotherapy could be beneficial, with consideration of citations from articles like PMID: 24480782, PMID: 28094262, PMID: 34094142, and PMID: 33643804.
Response: We are very grateful to the reviewer for recommending the good references. These papers have been cited in the manuscript (Reference [63], [64]).
- Typographical errors, such as 'et al.' in line 20, could be replaced with 'etc.'
Response: We thank the reviewer for helping us find the errors. The “et al.” has been replaced by “etc.” in revised manuscript.
- Additionally, typographical errors are observed in Line 119, Line 159, etc.
Response: We thank the reviewer for helping us point out the errors.
“This conclusion was confirmed by in vivo experiments in primary and distant tumor models, in which the Ce6-IMDQ treatment group was significantly superior to the other treatment groups in inhibiting tumor growth.” has been replaced by “The similar results were also obtained in primary and distant animal tumor models. Tumors growth was more effectively inhibited in the Ce6-IMDQ NPs treatment group compared to the other treatment groups.”
“AlPcE” has been replaced by “AlPc”
